# Crystallographic and Physicochemical Analysis of Bovine and Human Teeth Using X-ray Diffraction and Solid-State Nuclear Magnetic Resonance

**DOI:** 10.3390/jfb13040254

**Published:** 2022-11-19

**Authors:** Noriko Hiraishi, Tadamu Gondo, Yasushi Shimada, Robert Hill, Fumiaki Hayashi

**Affiliations:** 1Department of Cariology and Operative Dentistry, Graduate School of Medical and Dental Sciences, Tokyo Medical and Dental University, Tokyo 113-8549, Japan; 2Dental Physical Sciences Unit, Institute of Dentistry, Queen Mary University of London, London E1 4NS, UK; 3NMR Operation Team, Laboratory for Advanced NMR Application and Development, RIKEN Center for Biosystems Dynamics Research, Yokohama 230-0045, Japan

**Keywords:** enamel, dentin, apatites, X-ray diffraction, solid-state nuclear magnetic resonance, heteronuclear

## Abstract

Dental research often uses bovine teeth as a substitute for human teeth. The aim of this study was to evaluate differences in the crystalline nanostructures of enamel and dentin between bovine and human teeth, using X-ray diffraction (XRD) and solid-state nuclear magnetic resonance (NMR). The crystallite size (crystallinity) and microstrains were analyzed using XRD with the Rietveld refinement technique and the Halder–Wagner method. The ^31^P and ^1^H NMR chemical environments were analyzed by two-dimensional (2D) ^1^H-^31^P heteronuclear-correlation (HETCOR) magic-angle spinning (MAS) NMR spectroscopy. Enamel had a greater crystallite size and fewer microstrains than dentin for both bovine and human teeth. When compared between the species, the bovine apatite had a smaller crystallite size with more microstrains than the human apatite for both dentin and enamel. The 2D HETCOR spectra demonstrated that a water-rich layer and inorganic HPO_4_^−^ ions were abundant in dentin; meanwhile, the hydroxyl group in the lattice site was more dominant in enamel. A greater intensity of the hydroxyl group was detected in human than in bovine for both dentin and enamel. For ^31^P projections, bovine dentin and bovine enamel have wider linewidths than human dentin and human enamel, respectively. There are differences in the crystallite profile between human and bovine. The results of dental research should be interpreted with caution when bovine teeth are substituted for human teeth.

## 1. Introduction

In order to apply dental materials to dental practice, their physical, chemical, and biological properties must first be evaluated in vitro. In dental research, it is necessary to simulate human extracted teeth using a model. However, human teeth are not sufficient in quantity or quality. Researchers encounter infection hazards and ethical issues when extracted teeth are used for research purposes. Extracted teeth are generally more likely to reveal extensive carious lesions and restorations, except for the impacted third molars. Moreover, human teeth are prone to large variants in terms of their source and age, influencing the results among the selected tooth substrates [1]. Therefore, bovine teeth have been used as an excellent alternative to human teeth for in vitro dental research, although few studies have directly compared human and bovine hard tooth tissue.

In vitro and in situ studies have been conducted to compare the chemical composition [2,3], micro-morphology [4], and physical properties [5]. Several methods have been commonly used in fundamental studies to compare the properties of teeth obtained from human and other animal species by means of microradiography observation [6,7], polarized light microscopy [8], microhardness [8,9], and scanning electron microscopy [9,10,11]. However, applying more advanced analytical techniques is necessary to accurately interpret differences in the characteristics of bovine and human teeth.

Advanced research has been conducted on biological hard tissues using atomic force microscopy [12,13], Raman spectroscopy [14,15], XRD [14,16,17], and nuclear magnetic resonance (NMR) [18,19,20,21]. These techniques have been preliminarily employed to identify the microstructural characteristics of biological apatites, mainly for bone, but they have not been highlighted for comparing bovine and human teeth.

It is well known that biological apatite is nanocrystalline, structurally disordered, and compositionally nonstoichiometric [22]. Biological apatite has a high specific surface area [20,21], which contains labile chemical groups, ionic vacancies, and substitutions [22]. The surface’s physicochemical properties are essential to the interaction affinity of agents because the surface is composed of a water-rich hydrated layer, which contains easily exchangeable mobile ionic species [23] The crystallographic and physicochemical properties of bovine and human apatite should play an important role in the interpretation of dental investigation results, especially demineralization and remineralization phenomena. Therefore, these properties must be elucidated by definitive techniques that provide atomic-level detail.

A crystallographic nanostructure can be determined with an XRD measurement, followed by a Rietveld analysis, to theoretically estimate the lattice parameters. A detailed description of the mathematical procedures implemented in the Rietveld analysis has been provided in a previous study [24]. In brief, the Rietveld method enables a complete spectrum fitting to analyze X-ray and neutron diffraction data. This method uses lattice and structure parameters from X-ray and neutron powder diffraction data, from which structural information, such as fractional coordinates, occupancies, and atomic displacement parameters can be extracted.

The physicochemical properties of biological minerals can be demonstrated by solid-state magic-angle spinning (MAS) NMR spectroscopy. For human enamel and dentin, Kolmas et al. reported that the surface areas of those biological apatites were estimated by the linewidths of the ^31^P MAS NMR peaks [20]. Wu et al. employed a ^1^H-^31^P cross-polarization (CP) MAS NMR measurement and provided evidence related to bovine bone and enamel [18]. A significant fraction of the protonated phosphates (HPO4-2) was located on the biological crystals’ surfaces and the unprotonated phosphates (PO4-3) within the apatitic lattice.

Solid-state NMR (SS-NMR) spectroscopy also has the advantage of detecting hydroxyl (OH^−^) ions in biological calcium phosphates to confirm OH– content that is typical of synthetic hydroxyapatite [25]. However, 1D ^1^H MAS NMR spectra do not distinguish hydroxyl OH– in biological apatite in bone and dentin because of various resonances arising from protons in the organic matrix. To eliminate this influence, two-dimensional (2D) ^1^H-^31^P heteronuclear-correlation (HETCOR) MAS NMR spectroscopy is one of the most promising methods among the various NMR techniques. The 2D ^1^H-^31^P HETCOR suppresses the proton signal from the organic matrix in biological apatite and reveals the proton signals in hydrated calcium phosphate on the apatite’s mineral surface.

In our study, XRD with a Rietveld analysis and 2D ^1^H-^31^P HETCOR MAS NMR techniques were employed to compare the crystallographic and physicochemical properties of dentin and enamel samples obtained from bovine and human teeth. To the best of our knowledge, no studies comparing bovine and human teeth have been conducted using these techniques. 

## 2. Materials and Methods

### 2.1. Sample Preparation

Twelve bovine incisors and fifteen human non-caries third molars were used in this study. The bovine samples were aged less than 30 months and obtained as discarded specimens in authorized procedures that were approved by the Food Safety Commission of Japan, and the Ministry of Health, Labor and Welfare. Human teeth from 25 to 32 years of age (29.0 ± 3.2, means ± S.D.) were collected from the Japanese population following the guidance of the Ethical Committee at Tokyo Medical and Dental University under protocol number 725, D2013-022-03. The teeth were stored frozen and used within one month after extraction. After the soft tissues were removed, the dentin and enamel fragments were pulverized under liquid nitrogen into particles with diameters of less than 75 μm. The pulverized samples were dried to a constant weight at 37 °C and used for the XRD and SS-NMR measurements.

In addition, synthetic carbonate-free hydroxyapatite (syn-HAp) was prepared by the biomimetic precipitation method for comparison with natural biological apatite. The wet chemical technique was used to follow the methodology of Habraken et al. [26]. In brief, the precipitation reaction was performed in a Tris-buffered saline solution containing a 50 mM Trizma-base and 150 mM of sodium chloride (NaCl) in Milli-Q water, and the solution was set at pH 7.40. The apatite precipitation was accomplished by incubating 5.88 mM CaCl_2_ with 4.12 mM K_2_HPO_4_ at 37 °C for 24 h.

### 2.2. XRD Experiment for Crystallographic Structure

X-ray diffraction (XRD) (SmartLab, Rigaku Corporation, Tokyo, Japan) was equipped with a CuKα lamp (l = 1.5406 Å) operating at 45 kV and 200 mA. The XRD patterns were recorded between 8° and 100° on a 2θ scale in steps of 0.02° intervals with a counting time of 1.0° per min. Rietveld refinement theory has been previously described in the literature [24].

In the current study, the Rietveld analysis of the XRD patterns was performed by the Rietan-FP system [27,28,29]. Rietveld’s refinement was performed based on hexagonal space group P63/m (Hermann–Mauguin symbol); the number of the space group was 176. Isotropic displacement parameters (Uiso) were fixed to that of the syn-HAp results due to the difficulty of simultaneously refining the Uiso and fractional occupancies (Occ). The analysis proceeded with stoichiometric hydroxyapatite as the initial structure value (Appendix A), while all the values for the occupancy (Occ) of Ca were adjusted to 1.

The crystallite size and microstrains were determined by utilizing the Williamson–Hall and Halder–Wagner methods. Calculating crystallite sizes and microstrains was attempted by using the following equation:$${\left(\frac{\beta }{tan\theta }\right)}^{2}=\frac{K\lambda }{D}\cdot \frac{\beta }{tan\theta sin\theta }+16\epsilon^{2}$$
where *θ* is the Bragg angle, and *β*, *λ*, *D*, *ε*, and *K* are the integral breadth of the reciprocal lattice point, the wavelength of the X-ray or neutron beam, crystallites size, microstrain, and the shape factor, respectively. In the Halder–Wagner (HW) plot, $y={\left(\frac{\beta }{tan\theta }\right)}^{2}$ was used against $x=\frac{\beta }{tan\theta sin\theta }$. The slope and y-intercept of the linear line indicated $\frac{K\lambda }{D}$ and $16\epsilon^{2}$, respectively. The value of $K=\frac{3}{4}$ was used as the definition of the crystallite size for the volume-weighted average size for spherical crystallites [30]. The three dimensions of the crystal structures were visualized by means of the VESTA program [27,29].

### 2.3. Solid-State NMR Experiment for Physicochemical Properties

Solid-state NMR spectra were acquired with a JEOL ECAII 700 spectrometer (JEOL, Tokyo, Japan) (^1^H Larmor frequency of 700.625 MHz) with a JEOL 3.2 mm HX MAS probe. The powder samples were packed into cylindrical zirconia rotors with a 3.2 mm O.D. The 90° pulse length for ^1^H was 2.7 to 2.9 μs. The two-dimensional ^1^H-^31^P HETCOR spectra were acquired with a spinning speed of 15kHz and a contact time of 1 ms. A spin-lock pulse of between 4 and 10% was used on the ^31^P channel. A recycle delay of between 9 and 20 s, 12 to 48 transients per t1 time increment, and a 33.33 s time increment in the indirect dimension were used. The acquired matrix size was 128 complex points (t1) × 256 complex points (t2). ^1^H high-power decoupling was applied during the ^31^P acquisition. An 85% of the H_3_PO_4_ was used as an external ^31^P chemical shift reference (0 ppm). HAp was used as an external secondary chemical shift reference for ^31^P (2.9 ppm). 

A summation projection of the 2D HETCOR spectra was taken onto the ^31^P axis, and the linewidth of the ^31^P NMR band was considered at full width at half maximum (FWHM). For the summation projection onto the ^1^H axis, the intensity ratio of the two characteristic peaks was calculated. All the data processing was performed using the Delta^TM^ 5.1.3 NMR software (JEOL, Tokyo, Japan).

## 3. Results

### 3.1. XRD Experiment for Crystallographic Structure

The X-ray powder diffraction pattern of the samples is shown in Appendix A. The XRD patterns show the characteristic peaks of hydroxyapatite, which indicate the presence of a majority phase corresponding to the hexagonal symmetry-P63/m space group associated with HAp. In addition, no other minority phase was detected within the X-ray detection limits. 

Figure 1 shows the results of the Rietveld refinement of bovine dentin, bovine enamel, human dentin, and human enamel. The calculated and experimental diffraction profiles are shown together with the difference curve obtained after the final refinement. The reliability index parameters are indicated as weighted profile R-factor (Rwp), unweighted profile R factor (Rp), expected R factor (Re), and goodness-of-fit factor (GOF). The GOF value is established by comparing Rwp with the Re (Rwp/Re). Each of the reliability index parameters (Rwp) was sufficiently low, which indicates a good fit between the data calculated by the theoretical model and the observed pattern [31].

The lattice parameters are summarized in Table 1. The lattice parameter (a) was greater in enamel than dentin for both bovine and human teeth. When the lattice parameter (a) was compared between bovine and human teeth, there was no notable difference for dentin and enamel. The lattice parameter (c) was not remarkably different between the groups. This result indicates that the crystallite size in enamel was greater than that in dentin for both species; meanwhile, the difference in the crystallite size was not obvious between bovine and human teeth. 

Table 2 summarizes the occupancies and atomic coordinates. A lower occupancy of the Ca1 was revealed for dentin than enamel, but there was no remarkable difference within species. There are differences in the occupancies of the P site between dentin and enamel, which shows lower occupancies of the P in dentin than in enamel for both species. When compared between bovine and human dentin, it is noteworthy that the occupancy of P is lower for the former. Meanwhile, for enamel, the occupancies Ca1 and P are greater for bovine enamel than human enamel. The atomic coordinate of O4 site, i.e., O(H), is 0.072, 0.102, 0.167, and 0176 for bovine dentin, human dentin, bovine enamel, and human enamel, respectively. Since the site of the O4 of stoichiometric hydroxyapatite is 0.18819 (Appendix A), this indicates that the hydroxyl in dentin for both species, especially for bovine dentin, is located proximately to the (0,0,0) site compared to that of stoichiometric hydroxyapatite. The displacement of O4 site is visualized by the VESTA system in Figure 2.

The Halder–Wagner plots are shown in Appendix A, and the crystallite sizes and microstrains analyzed by means of the Halder–Wagner method are collected in Table 3. The dentin is smaller in crystallite size with more strains than the enamel. For both dentin and enamel, bovine teeth show a smaller crystallite size and greater microstrains than human teeth. 

### 3.2. Solid-State NMR Experiment for Physicochemical Properties

Figure 3 shows the 2D ^1^H-^31^P HETCOR spectra of the dentin (Figure 3A) and the enamel (Figure 3B) to compare the bovine and human samples. The spectra revealed two prominent cross-peaks for dentin and prominent cross-peaks for the enamel samples.

On the ^1^H axis, two ^1^H peaks were present at around 0.5 ppm and 6 ppm. The downfield peak at 6 ppm was pronounced for the dentin samples and tailed downfield (downward) to 18 ppm. According to previous NMR studies, the peak at around 0.5 ppm was assigned to hydroxyl ions (OH^−^) in the apatite lattice site; meanwhile, the downfield peak at around 6.0 ppm to 18 ppm was attributed to proton signals in the disordered surface layer [32]. To be accurate, the signal at the 6 ppm peak represents surface water, and the greater downfield signal to 18 ppm represents inorganic HPO_4_^2−^ ions, which are also abundant on the apatite surface. At the summation projection onto the ^31^P axis, the dentin and enamel sample presented a symmetric line shape. The enamel showed sharp peaks, and the dentin showed a relatively broad peak. The FWHMs of both dentin and enamel are smaller for human than for bovine. The ^1^H peak at around 0.5 ppm, hydroxyl ions (OH^−^), was more prominent for enamel than dentin. The intensity ratio of the two characteristic ^1^H peaks, expressed as [core around 0.5 ppm]/[surface around 6.0 ppm], showed a greater ratio in human than in bovine for both dentin and enamel. The cross-peaks, FWHM, and ^1^H intensity ratio results are summarized in Table 4.

## 4. Discussion

In this study, the use of Rietveld refinement enabled the extraction of information about s-unit dimensions, crystallite sizes, and microstrain in the crystal lattice. A notable finding was that the occupancies of the P atom were lower for dentin than enamel when compared with the stoichiometric hydroxyapatite model, especially for bovine dentin. The fact that P atoms show vacancies may indicate evidence of the replacement of PO_4_^3−^ by CO_3_^2−^ ions in the apatite lattice. In terms of Ca occupancy, some sites were less than 1, and small variations were observed, whereas the lattice parameter and occupancies did not vary significantly between species. The replacement of the Ca site occurs when Ca^2+^ is substituted for other cations, such as Mg^2+^, Sr^2+^, and Zn^2+^ [33]. Carbonate apatite, especially type B carbonate apatite (substitution of CO_3_^2−^ for PO_4_^3−^), is also considered to contribute to Ca deficiency. This is due to other cations replacing Ca^2+^ to maintain the charge balance after PO_4_^3−^ loss [18,33].

The substitution of atoms in the crystal lattice may cause changes in the crystal lattice parameter. Bazin et al. addressed that pure type A substitution (substitution of CO_3_^2−^ for OH^−^) induces enlargement of the crystallographic parameter a, and a reduction in the crystallographic parameter c [33]. On the other hand, pure type B carbonate apatite causes a contraction of the a parameter and an expansion of the c parameter. Contrary to expectations, the present results showed no remarkable differences in the a and c lattice parameters. This unexpected result may be attributed to an imperfect model for structure refinement.

The minuscule difference was visualized by VESTA to reflect the atomic coordinates and occupancies ratio (Figure 3). Interestingly, the atomic coordinates (z) of the O4 site, i.e., O(H), exhibited that the hydroxyl groups in the dentin samples were located proximately to the (0,0,0) site on the c-axis. This is more prominent for bovine dentin compared with human dentin. The reason for the different atomic coordinates (z) may be related to a substitution of OH^−^ ion by another monovalent ion, e.g., Cl^−^ or F^−^ [34]. Indeed, fluoride, which is commonly found in dentin and enamel, is thought to substitute OH^−^ in the apatite during the development phase of dentin and enamel [35].

Human dentin and enamel presented greater crystallite sizes and reduced microstrain compared to bovine dentin and enamel. A higher microstrain parameter means lower crystallinity and a larger specific sample surface area (Yan et al. 2013).The crystal surface of biological apatite is remarkably heterogeneous and covered with water and various adsorbed ions (Combes et al. 2016). The crystal surfaces play an essential role in the physiological activity, mediating ion exchange and modulating their adsorption capacity. Resistance, reactivity, and solubility to aggressive substances, such as acids, are influenced primarily by surface properties (Cazalbou et al. 2005). Therefore, for the pharmacological effects used in dentistry, it is essential to characterize the crystal surfaces of dentin and enamel.

The SS-NMR experiment also indicated the differential properties between the enamel and dentin samples. The ^31^P NMR spectra are broader in dentin than in enamel, which means that the phosphates in dentin tend to be in a more disordered environment. The spectral linewidth of ^31^P was narrower for enamel than dentin, which is consistent with the previous report [20]. The linewidths of the dentin and enamel of human were narrower than those of bovine. Kolmas et al. reported that the linewidth of HAp decreased with an increasing calcination temperature, resulting in a more ordered crystal lattice, and thus, a larger average crystal size [20]. This explanation for the narrower linewidth indicated that human dentin and enamel have a more ordered crystal lattice and larger average crystal size than bovine dentin and enamel, respectively. This result was consistent with that of the Rietveld refinement analysis.

It may be noted that octacalcium phosphate (OCP), Ca_8_(HPO_4_)_2_ (PO_4_)_4_. 5H_2_O, may be one of the precursors to apatite crystallites [36]. Previously, the presence of OCT was suggested by the increased FWHM detected in human teeth compared to HAp [20]. The NMR shift of authentic OCP is characterized by a very distinct ^31^P profile, with peaks at -0.3, 2, 3.2, and 3.5 ppm [37]. However, the current NMR results did not provide conclusive evidence of OCPs, probably because the dentin and enamel samples did not contain significant amounts of OCP. For XRD, the characteristic XRD pattern of OCP has a strong (100) peak at 4.73° (Lu et al. 2000), which was not obtained in our XRD patterns recorded between 8° and 100°. Nevertheless, the possibility that dentin or enamel samples may contain remnants of degraded OCP cannot be ruled out, which might influence the current XRD and NMR results.

In order to compare the apatitic differences between bovine and human teeth, we focused on the characteristics of the crystalline lattice and surface regions. The relatively sharp peaks ranging from 0.48 to 0.65 ppm were likely assigned as a crystalline core. Meanwhile, the broad peaks ranging from 5.67 to 6.59 were presumed to be surface water. The ^1^H [38], ^2^H NMR [39], and 2D ^1^H-^31^P HETCOR [40,41] experiments revealed that the broad peak corresponded to protons from either adsorbed or structural water. Yesinowski and Eckert identified surface-adsorbed water at the 5.6 ppm peak because this peak was not observed with lower surface areas and its intensity decreased upon exposure to D_2_O (deuterium oxide) [38]. Wilson et al. suggested a very ordered structural layer of water molecules that hydrates the small bioapatite crystallites in bone through very close arrangements [42]. It is difficult to confirm the abundance of structural water in our study, but Table 4 shows that the intensity ratio of the two characteristic ^1^H peaks, expressed as [core around 0.5 ppm]/[surface around 6.0 ppm], was greater in enamel than dentin. The ratio was greater in human samples than in bovine samples. The results indicate that dentin, especially bovine dentin, contains a considerable hydrated mineral fraction.

A previous ^31^P SS-NMR study examined HPO_4_^−2^ ions in enamel obtained from 4-week-old to 4-year-old bovine teeth, and the data were compared with those obtained from pure and carbonated hydroxyapatites [18]. Wu et al. provided evidence that a significant portion of protonated phosphate (HPO_4_^2−^) was located on the surfaces of the biological crystals, and the total concentration of the surface HPO_4_^2−^ groups was higher in the younger, less mature biological crystals. They indicated that the concentration of phosphate vacancies in the internal lattice of bioapatite crystals decreases as the enamel ages [18]. In the current results, a broad peak at 6 ppm tailed off to around 18 ppm for dentin, suggesting the presence of HPO_4_^2−^. The result indicates that the phosphate vacancies in bovine dentin were relatively greater than in human dentin.

The differences in the crystal size, microstrain, and surface properties of bovine and human bioapatite may be attributed to the age of the crystals. The bovine teeth were obtained from relatively young ages, less than 30 months; meanwhile, the human teeth were aged from 25 to 32 years old. For human teeth, the initial calcification of the third molars occurs between 7 and 10 years, and eruption time between 17 and 21 years [43]. Mineralization occurs until the root is completed at 18–25 years [43]. Thus, the crystals of human teeth were relatively mature when the extracted teeth were collected. 

Differences in the developmental stage of the teeth partially contribute to the differences in crystallographic and physicochemical properties. To eliminate differences in the crystallization stage, it is essential to use bovine teeth with mature crystallization, similar to human teeth. However, cattle older than 30 months are considered at high risk for bovine spongiform encephalopathy [44], and it is difficult to obtain mature bovine teeth. 

In general, bioapatite shows a greater specific surface area than stoichiometric HAp [18]. The smaller the apatite crystals, the larger the specific surface area. The apatite crystal surface serves as an interface for chemical agents such as ions, proteins, and drugs, in order to achieve their sorption [22]. The difference between the crystal surface of bovine and human teeth must be considered when interpreting results obtained from any experiment with bovine teeth substrate. Moreover, micro-morphological, chemical composition, and physical property differences should be responsible for inconsistent data regarding whether bovine teeth can be considered an appropriate substitute for human teeth in dental research [45]. 

## 5. Conclusions

Within the limitations of the present study, the differences in crystallographic and physicochemical properties were demonstrated between dentin and enamel, as well as between bovine and human origin. Dentin had a smaller crystal size (i.e., a large surface area) with greater microstrains (i.e., ions vacancies and disorders) than enamel. The surface layer was more obviously present around the crystal core in dentin than in enamel. When compared between species, bovine teeth indicate a smaller crystal size, more microstrains, and a greater surface layer than human teeth. The results of dental research should be interpreted with caution when bovine teeth are substituted for human teeth. 

## Figures and Tables

**Figure 1 jfb-13-00254-f001:**
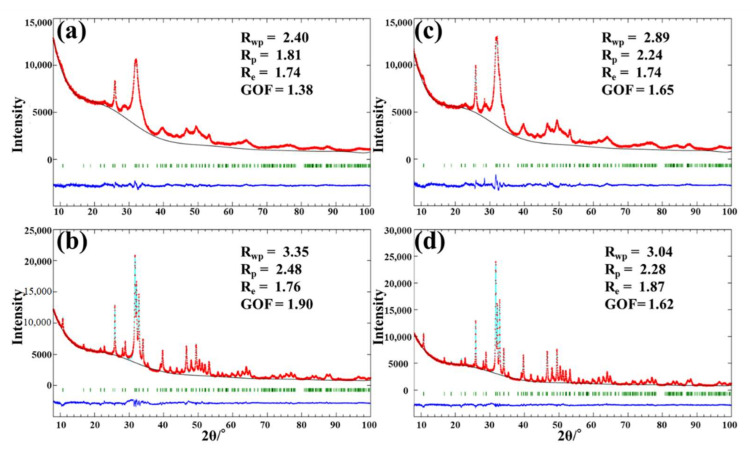
The results of the Rietveld refinement of (**a**) bovine dentin, (**b**) bovine enamel, (**c**) human dentin, and (**d**) human enamel. The calculated pattern (solid light blue) and observed diffraction (doted red line) profiles are shown, as well as the difference line obtained after the final refinement. The small trace in the difference line indicates the good agreement between the calculated and measured diffraction profile. Rwp, weighted profile R-factor; Rp, unweighted profile R factor; Re, expected R factor; and GOF, goodness-of-fit factor. Based on the Rwp values, the Rietveld refinement fittings of the X-ray data were good for all the samples. The lower trace is the difference between observed and calculated patterns, and the vertical lines mark the positions of the calculated Bragg peaks.

**Figure 2 jfb-13-00254-f002:**
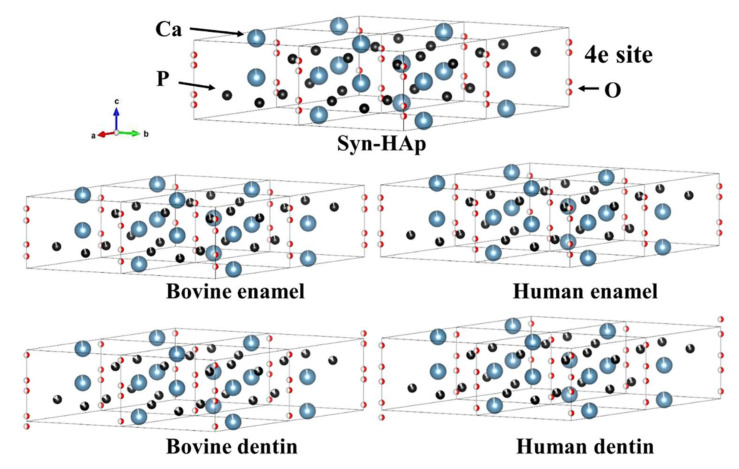
The polyhedral model using three-dimensional visualization program VESTA. Note: the representations of Ca (blue), P (black), O (red).

**Figure 3 jfb-13-00254-f003:**
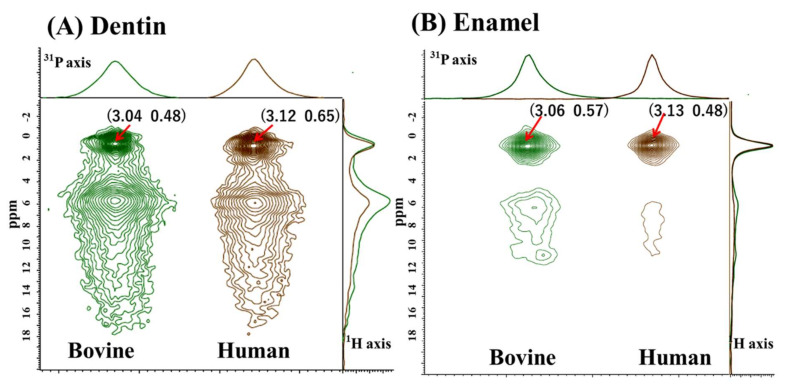
Two-dimensional ^1^H-^31^P HETCOR of (**A**) dentin and (**B**) enamel. The summation projection to the ^31^P axis is on the upper side of the HETCOR spectrum, and the summation projection to the ^1^H axis is on the right. On the ^31^P axis, the ^31^P spectra are normalized to maximum intensities. On the ^1^H axis, the ^1^H spectra are normalized to the intensities of the up-field shift at around 0.5 ppm.

**Table 1 jfb-13-00254-t001:** Lattice parameters of bovine dentin, bovine enamel, human dentin, and human enamel.

Lattice Parameters	a/Å	b/Å	c/Å
Bovine dentin	9.425	=a	6.8735
Bovine enamel	9.4366	=a	6.8798
Human dentin	9.4250	=a	6.8789
Human enamel	9.4374	=a	6.8780
Syn-HAp	9.4569	=a	6.8752

Note: a hexagonal structure with a P63/m space group and cell dimensions a = b = 9.42 Å and c = 6.88 Å.

**Table 2 jfb-13-00254-t002:** Atomic coordinates and occupancies for each sample.

	Bovine Dentin					Bovine Enamel				
Site	Occ	x	y	z	Uiso	Occ	x	y	z	Uiso
Ca1	0.949	1/3	2/3	0.0106	0.0036	0.950	1/3	2/3	0.0096	0.0036
Ca2	1	0.2344	0.9840	¼	0.0082	1	0.2386	0.9857	¼	0.0082
P	0.811	0.4072	0.3785	¼	0.00253	0.872	0.4020	0.3754	¼	0.00253
O1	1	0.351	0.507	¼	0.011	1	0.351	0.505	¼	0.011
O2	1	0.5926	0.456	¼	0.011	1	0.5877	0.457	¼	0.011
O3	1	0.3437	0.2668	0.0734	0.011	1	0.3468	0.2652	0.0721	0.011
O4	0.5	0	0	0.072	0.005	0.5	0	0	0.102	0.005
Human dentin					Human enamel					
Site	Occ	x	y	z	Uiso	Occ	x	y	z	Uiso
Ca1	0.983	1/3	2/3	0.0042	0.0036	0.969	1/3	2/3	0.0029	0.0036
Ca2	1	0.2478	0.9916	¼	0.0082	0.969	0.2482	0.9918	¼	0.0082
P	0.945	0.3974	0.3674	¼	0.00253	0.916	0.3981	0.3682	¼	0.00253
O1	1	0.3357	0.4886	¼	0.011	1	0.3322	0.4858	¼	0.011
O2	1	0.5833	0.4647	¼	0.011	1	0.5840	0.4634	¼	0.011
O3	1	0.3441	0.2630	0.0672	0.011	1	0.3412	0.2603	0.0701	0.011
O4	0.5	0	0	0.167	0.005	0.5	0	0	0.176	0.005
Syn-HAp										
Site	Occ	x	y	z	Uiso					
Ca1	0.979	1/3	2/3	0.0022	0.0036					
Ca2	0.978	0.2458	0.990	¼	0.0082					
P	1	0.3946	0.3687	¼	0.00253					
O1	1	0.3399	0.4949	¼	0.011					
O2	1	0.5859	0.4644	¼	0.011					
O3	1	0.3414	0.2571	0.0733	0.011					
O 4	0.5	0	0	0.186	0.005					

**Table 3 jfb-13-00254-t003:** The crystallite sizes and microstrains of bovine dentin, bovine enamel, human dentin, and human enamel, as analyzed by the Halder–Wagner method.

	Crystallite Size (nm)	Microstrains (%)
Bovine dentin	5.5	0.90%
Bovine enamel	22.1	0.14%
Human dentin	5.9	0.25%
Human enamel	36.7	0.08%

**Table 4 jfb-13-00254-t004:** Summary of SS-NMR analysis.

	Bovine Dentin	Bovine Enamel	Human Dentin	Human Enamel
^31^P cross-peak at 2D HETCOR (ppm)	3.04	3.06	3.12	3.13
FWHM on ^31^P axis (kHz)	1.31	0.72	1.16	0.54
Core peak on ^1^H axis (ppm)	0.48	0.57	0.65	0.48
Surface peak on ^1^H axis (ppm)	5.67	5.92	5.92	6.59
Intensity ratio of [core 0.5 ppm]/[surface 6 ppm]	0.67	6.38	1.19	13.57

## Data Availability

Not applicable.

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
