# Peer review of "Crystallographic and Physicochemical Analysis of Bovine and Human Teeth Using X-ray Diffraction and Solid-State Nuclear Magnetic Resonance"

_jfb, 2022, doi:10.3390/jfb13040254_

Round 1

Reviewer 1 Report

This paper can be published a minor revison. Please, check and correct page 4, line 131.

Author Response

Response to the comment: The line has been corrected.

Reviewer 2 Report

1. The authors should have maintained the even size of the letters throughout the manuscript.

2. The tables should be arranged appropriately with their titles highlighted accordingly.

3. Table 1 has been added twice in the manuscript. 

4. Kindly arrange the manuscript according to the journal standard.

Author Response

1. The authors should have maintained the even size of the letters throughout the manuscript.

Response to the comment: The line has been corrected.

2. The tables should be arranged appropriately with their titles highlighted accordingly.

Response to the comment: The tables have been arranged correctly.

3. Table 1 has been added twice in the manuscript. 

Response to the comment: Table 1 has been corrected.

4. Kindly arrange the manuscript according to the journal standard.

Response to the comment: The manuscript has been corrected.

Reviewer 3 Report

This is an interesting piece of work involving crystallographic and other syudeis with bovine and human teeth. This work has been generally carried out well so I recommend its publication. However, there are a couple of concerns that should be addressed. 

1) The Introduction section needs substantial improvement. It's not all lucid and anyone who is not directly involved in this particular research area will have a hard time following this smoothly, so I highly recommend the authors rewrite the Introduction section.

2) There are sentence construction errors in several places in the manuscript. All those issues should be fixed prior to final approval for publication.

Author Response

1) The Introduction section needs substantial improvement. It's not all lucid and anyone who is not directly involved in this particular research area will have a hard time following this smoothly, so I highly recommend the authors rewrite the Introduction section.

Response to the comment: The Introduction has been revised as follows:

In order to apply dental materials to dental practice, their physical, chemical, and biological properties must first be evaluated in vitro. In dental research, it is necessary to simulate human extracted teeth as a model. However, human teeth are not sufficient in quantity or quality.

2) There are sentence construction errors in several places in the manuscript. All those issues should be fixed prior to final approval for publication.

Response to the comment: The layout has been fixed, and the manuscript has been edited by English Editing service recommended by MDPI. 

Round 2

Reviewer 2 Report

The authors have made all the correction as per the previous comments.

Reviewer 3 Report

I am happy with the present form of the manuscript and I recommend its publication.